# Generative acceleration of molecular dynamics simulations for solid-state electrolytes

**Juno Nam** [1]  **Sulin Liu** [1]  **Gavin Winter** [1]  **Rafael Gómez-Bombarelli** [1]

## Abstract

We introduce LIFLOW, a generative acceleration framework designed for efficiently simulating diffusive dynamics in solids, particularly lithium-based solid-state electrolytes (SSEs). LIFLOW consists of two components: *Propagator* and *Corrector*, which utilize a conditional flow matching scheme to predict atomic displacements and perform denoising, respectively. Our model achieves a Spearman's rank correlation of approximately 0.7 for the lithium mean squared displacement (MSD) on test set based on composition and temperature splits and offers a substantial speedup compared to reference molecular dynamics (MD) simulations using machine learning interatomic potentials (MLIPs). This framework facilitates high-throughput virtual screening for electrolyte materials and holds promise for the optimization of the kinetic properties of crystalline solids.

## 1. Introduction

Solid-state electrolytes (SSEs) are an emerging class of materials for electrical energy storage, offering a safer and more stable alternative to the liquid electrolytes commonly used in lithium-ion batteries (Bachman et al., 2016). The study and design of SSEs require fast and accurate atomistic simulation techniques to model the intricate ionic diffusion behaviors in such materials. The standard method, *ab initio* molecular dynamics (AIMD), involves costly density functional theory (DFT) calculations for each propagation step. Hence, their application is limited to small spatiotemporal scales and a few simulations, often insufficient for characterizing diffusive dynamics or screening candidate materials. Recently, universal machine learning interatomic potentials (MLIPs), trained on large-scale DFT calculations,

[1]Department of Materials Science and Engineering, Massachusetts Institute of Technology, Cambridge, MA 02139, USA. Correspondence to: Rafael Gómez-Bombarelli <rafagb@mit.edu>.

*Accepted at the 1st Machine Learning for Life and Material Sciences Workshop at ICML 2024.* Copyright 2024 by the author(s).

have emerged as a promising alternative (Friederich et al., 2021; Ko & Ong, 2023). Still, even with MLIPs, dynamics must be discretized at sufficiently small time steps to ensure stable and accurate propagation (Fu et al., 2023a), limiting the scalable analysis of large-scale materials databases.

In the context of general MD simulations, methods such as Timewarp (Klein et al., 2024), Implicit Transfer Operator Learning (Schreiner et al., 2024), and Score Dynamics (Hsu et al., 2024) have been proposed to tackle the challenge of accelerating the simulation. These methods leverage a generative modeling framework to propagate the conformational distribution from time $\tau$ to time $\tau + \Delta\tau$, where $\Delta\tau$ is much larger than the typical MD time steps. A similar approach has been applied to temporally coarse-graining polymer electrolyte simulations (Fu et al., 2023b).

Building upon these methods, this work aims to develop a tailored generative modeling framework, specifically designed for cost-effective simulation of diffusive dynamics in SSEs. Our primary objective is to construct a model that accurately reproduces relevant kinetic observables, such as mean squared displacement (MSD) and diffusivity of mobile ions, compared to long-time MD simulations utilizing MLIPs. Preferably, the model would exhibit chemical (encompassing electrolytes with diverse elemental compositions) and thermal (spanning various simulation temperatures) transferability, thereby offering a generalizable enhancement in simulation efficiency.

## 2. Methods

### 2.1. Overview

The scheme for LIFLOW is depicted in Fig. 1. LIFLOW models the distribution of atom positions in a system with a periodic boundary at time $\tau + \Delta\tau$ given the positions at time $\tau$. Here, we fix the $\Delta\tau$ to 1 ps, which is $10^3$ times larger than the MD time step $\delta\tau = 1$ fs. Inspired by Fu et al. (2023b), LIFLOW consist of two modules, *Propagator* and *Corrector*, which are both flow matching generative models that generate a displacement vector for each atom in a system. Given atom positions $\mathbf{x}_\tau$ at time $\tau$, *Propagator* samples a displacement vector $\mathbf{d}^P = \tilde{\mathbf{x}}_{\tau+\Delta\tau} - \mathbf{x}_\tau$. The displacement is then added to $\mathbf{x}_\tau$ to give a candidate atom positions $\tilde{\mathbf{x}}_{\tau+\Delta\tau}$ at

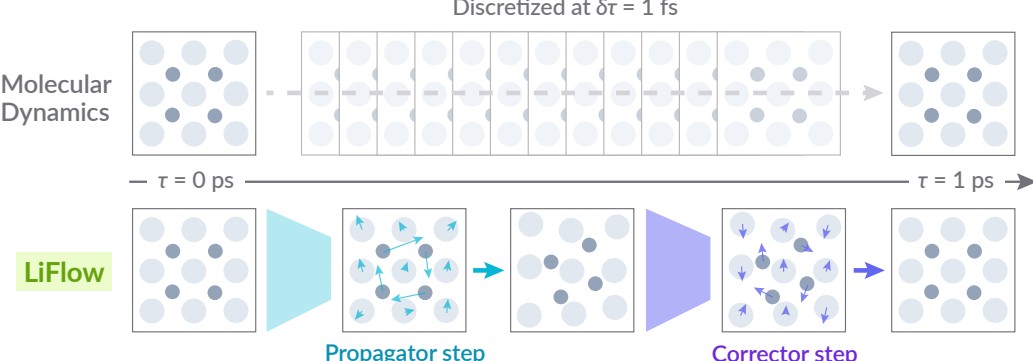

*Figure 1.* **LiFlow scheme.** LiFlow is a generative acceleration framework for SSE MD simulations, with *Propagator* and *Corrector* components leveraging a conditional flow matching scheme for accurate prediction of atomic displacements during time propagation.

the next time step. Since the displacements generated by *Propagator* model may not be accurate, we introduce an additional generative model, *Corrector*, which samples a displacement vector $\mathbf{d}^C = \mathbf{x}_{\tau+\Delta\tau} - \tilde{\mathbf{x}}_{\tau+\Delta\tau}$ to "denoise" or correct the atom positions. Similarly, the displacement is added to $\tilde{\mathbf{x}}_{\tau+\Delta\tau}$ to give a final prediction of the new atom positions $\mathbf{x}_{\tau+\Delta\tau}$.

**Notations** We denote the physical time by $\tau \in \mathbb{R}_{\geq 0}$ and flow matching time by $t \in [0, 1]$, which are associated with atomic positions $\mathbf{x}_\tau \in \mathbb{R}^{N \times 3}$ and displacements $\mathbf{d}_t^P, \mathbf{d}_t^C \in \mathbb{R}^{N \times 3}$ from the *Propagator* and *Corrector*, respectively. To featurize the input graph and propagate the dynamics, we additionally utilize lattice array $L \in \mathbb{R}^{3 \times 3}$ and atom types $\mathbf{a} \in \mathbb{N}^N$, atomic masses $\mathbf{m} \in \mathbb{R}_{>0}^N$ and the temperature $T \in \mathbb{R}_{\geq 0}$. At inference time, we discretize the flow matching generation into $N_{\text{flow}}$ steps, and iterate the LiFlow inference $N_{\text{step}}$ times to generate the final atom positions at time $N_{\text{step}}\Delta\tau$.

### 2.2. Conditional flow matching models for time propagation and correction

**Conditional flow matching.** Flow matching (Lipman et al., 2023; Tong et al., 2024) is a generative modeling framework in which samples from the prior distribution $\mathbf{d}_0 \sim p_0(\mathbf{d})$ is transported to samples from the data distribution $\mathbf{d}_1 \sim p_1(\mathbf{d})$ by a time-dependent vector field $u_t(\mathbf{d})$. Conditional flow matching render the vector field tractable by conditioning the time-dependent probability path and the vector field on the data, i.e., $p_t(\mathbf{d}|\mathbf{d}_1)$ and $u_t(\mathbf{d}|\mathbf{d}_1)$, respectively. The marginal vector field model $v_t(\mathbf{d}; \theta)$ is parametrized by a neural network and learned by the following regression objective:

$$\mathcal{L}_{\text{CFM}}(\theta) = \mathbb{E}_{t, \mathbf{d}_1, \mathbf{d}} \|v_t(\mathbf{d}; \theta) - u_t(\mathbf{d}|\mathbf{d}_1)\|^2, \quad (1)$$

where $t \sim \mathcal{U}[0, 1]$, $\mathbf{d}_1 \sim p_1(\mathbf{d})$, and $\mathbf{d} \sim p_t(\mathbf{d}|\mathbf{d}_1)$. Following Jing et al. (2024), we choose the linear interpolation

between the prior sample and the data sample as a conditional flow, i.e.,

$$u_t(\mathbf{d}|\mathbf{d}_1) = (\mathbf{d}_1 - \mathbf{d})/(1 - t). \quad (2)$$

Accordingly, we can alternatively predict $\mathbf{d}_1$ instead of the vector field $v_t$. By defining a final displacement predictor $\hat{\mathbf{d}}_1(\mathbf{d}, t; \theta)$, we parametrize the marginal vector field by

$$v_t(\mathbf{d}; \theta) = (\hat{\mathbf{d}}_1(\mathbf{d}, t; \theta) - \mathbf{d})/(1 - t). \quad (3)$$

The *Propagator* and *Corrector* in our scheme are hence defined as "$\mathbf{d}_1$ predictors," where we predict the desired displacements for propagation and correction and interpolate with the current displacement in each flow matching step.

**Maxwell–Boltzmann prior.** We design the LiFlow model to be transferable over different temperatures by introducing the temperature conditioning into the prior distribution of flow matching schemes. We use a scaled Maxwell–Boltzmann distribution as our prior, i.e., $\mathbf{d}_0 \sim \mathcal{N}(\mathbf{0}, (k_{\text{B}}T/\mathbf{m})\sigma^2)$, where $k_{\text{B}}$ is the Boltzmann constant and $\sigma$ is a hyperparameter for the model. This choice is motivated by the observation that atoms in the system diffuse faster when the particle masses are lighter and the temperature is higher. Note that the propagation time step $\Delta\tau$ is much larger than the velocity decorrelation time, so the direct physical relationship to the thermal velocities is not retained.

### 2.3. Model architecture

The input to the *Propagator* and *Corrector* models are current atomic positions $\mathbf{x}_\tau$ (or $\tilde{\mathbf{x}}_\tau$), current displacements $\mathbf{d}_t$, current flow matching time $t$, and atomic identities $\mathbf{a}$. Given these inputs, the models should predict the desired displacement $\mathbf{d}$. Since the atomic positions $\mathbf{x}_\tau$ represent their equivalent periodic images $\mathbf{x}_\tau + \mathbf{k}L^\top$ ($\mathbf{k} \in \mathbb{Z}^{N \times 3}$), the output displacements should be *invariant* to the periodic

translation of each atom. We represent the periodic atomic system as a graph with a fixed edge cutoff distance, and use PaiNN (Schütt et al., 2021) architecture to predict the displacements. The atomic identities $\mathbf{a}$ and flow matching time $t$ are embedded as node scalar features, and current displacements $\mathbf{d}_t$ are given as node vector features. The model output layer (gated equivariant block) is modified to predict a single vector for each node, which correspond to the final displacement prediction $\hat{\mathbf{d}}_1$.

### 2.4. Training and inference

*Propagator* **training.** The *Propagator* model is trained on the time-separated pairs of atom positions in a MD trajectory, $(\mathbf{x}_\tau, \mathbf{x}_{\tau+\Delta\tau})$. We obtain $\mathbf{d}_t^P$ from the interpolation between a sampled prior displacement $\mathbf{d}_0^P$ and the true displacement $\mathbf{d}_1^P = \mathbf{x}_{\tau+\Delta\tau} - \mathbf{x}_\tau$, according to eq. 2. Then, the $L_2$ loss between the model prediction $\hat{\mathbf{d}}_1^P = Propagator(\mathbf{x}_\tau, \mathbf{d}_t^P, L, \mathbf{a}, t)$ and the true displacement $\mathbf{d}_1$ is minimized.

*Corrector* **training.** For atom position $\mathbf{x}_\tau$ sampled in a MD trajectory, we add a noise $\boldsymbol{\epsilon} \sim \mathcal{N}(\mathbf{0}, \sigma^2)$ where $\sigma$ is sampled from $\mathcal{U}[0, 0.75]$ (units in Å) to obtain a noisy position $\tilde{\mathbf{x}}_\tau = \mathbf{x}_\tau + \boldsymbol{\epsilon}$. To address potential instabilities at very short distances, we resolve collisions by separating atom pairs within a cutoff distance (0.3 Å) along their pairwise directions until all pairs exceed the cutoff distance. Then, the *Corrector* model is trained similarly to the *Propagator*, with the model prediction obtained as $\hat{\mathbf{d}}_1^C = Corrector(\tilde{\mathbf{x}}_\tau, \mathbf{d}_t^C, L, \mathbf{a}, t)$ and the true displacement $\mathbf{d}_1^C = -\boldsymbol{\epsilon}$.

**LIFLOW inference.** Given the initial atom positions $\mathbf{x}_0$, we alternate between the *Propagator* and *Corrector* inference for $N_{\text{step}}$ steps to predict the final positions $\mathbf{x}_\tau$ at time $\tau = N_{\text{step}}\Delta\tau$. The inference for two flow matching models start with sampling a prior displacement $\mathbf{d}_0$ from Maxwell–Boltzmann distribution, predicting final displacement $\hat{\mathbf{d}}_1$, and interpolating with the current displacement to follow the marginal vector field in eq. 3. The flow is discretized to $N_{\text{flow}} = 10$ steps at inference time. The detailed inference algorithm is shown in Alg. 1.

## 3. Experiments

### 3.1. Datasets

To train a compositionally transferable generative model for time-shifting conformational distributions, we require long-time simulation trajectories that encompass diverse compositional spaces of solid-state materials. We fetched 4,186 lithium-containing structures from Materials Project (Jain et al., 2013) with the criteria of (1) more than 10% of the atoms are lithium, (2) band gap > 2 eV, and (3) energy

---

**Algorithm 1:** LIFLOW Inference

**Input:** Initial position $\mathbf{x}_0$, lattice $L$, atom types $\mathbf{a}$,
   atomic masses $\mathbf{m}$, temperature $T$
**Output:** Predicted position $\mathbf{x}_\tau$ at $\tau = N_{\text{step}}\Delta\tau$

**for** $i_\tau \leftarrow 0$ **to** $N_{\text{step}} - 1$ **do**
  $\quad \tau \leftarrow i_\tau \Delta\tau$ and $\tau' \leftarrow (i_\tau + 1)\Delta\tau$
  $\quad$ Sample $\mathbf{d}_0^P \sim \mathcal{N}(\mathbf{0}, (k_B T/\mathbf{m})\sigma_P^2)$
  $\quad$ **for** $i \leftarrow 0$ **to** $N_{\text{flow}} - 1$ **do**
    $\quad\quad t \leftarrow i/N_{\text{flow}}$ and $t' \leftarrow (i+1)/N_{\text{flow}}$
    $\quad\quad \hat{\mathbf{d}}_1^P \leftarrow Propagator(\mathbf{x}_\tau, \mathbf{d}_t^P, L, \mathbf{a}, t)$
    $\quad\quad \mathbf{d}_{t'}^P \leftarrow \mathbf{d}_t^P + (\hat{\mathbf{d}}_1^P - \mathbf{d}_t^P)/(1-t)N_{\text{flow}}$
    $\quad\quad$ (eq. 3)
  $\quad \tilde{\mathbf{x}}_{\tau'} \leftarrow \mathbf{x}_\tau + \mathbf{d}_1^P$ /* Propagated x */
  $\quad \tilde{\mathbf{x}}_{\tau'} \leftarrow$ ResolveCollision$(\tilde{\mathbf{x}}_{\tau'})$
  $\quad$ Sample $\mathbf{d}_0^C \sim \mathcal{N}(\mathbf{0}, (k_B T/\mathbf{m})\sigma_C^2)$
  $\quad$ **for** $i \leftarrow 0$ **to** $N_{\text{flow}} - 1$ **do**
    $\quad\quad t \leftarrow i/N_{\text{flow}}$ and $t' \leftarrow (i+1)/N_{\text{flow}}$
    $\quad\quad \hat{\mathbf{d}}_1^C \leftarrow Corrector(\tilde{\mathbf{x}}_{\tau'}, \mathbf{d}_t^C, L, \mathbf{a}, t)$
    $\quad\quad \mathbf{d}_{t'}^C \leftarrow \mathbf{d}_t^C + (\hat{\mathbf{d}}_1^C - \mathbf{d}_t^C)/(1-t)N_{\text{flow}}$
    $\quad\quad$ (eq. 3)
  $\quad \mathbf{x}_{\tau'} \leftarrow \tilde{\mathbf{x}}_{\tau'} + \mathbf{d}_1^C$ /* Corrected x */

---

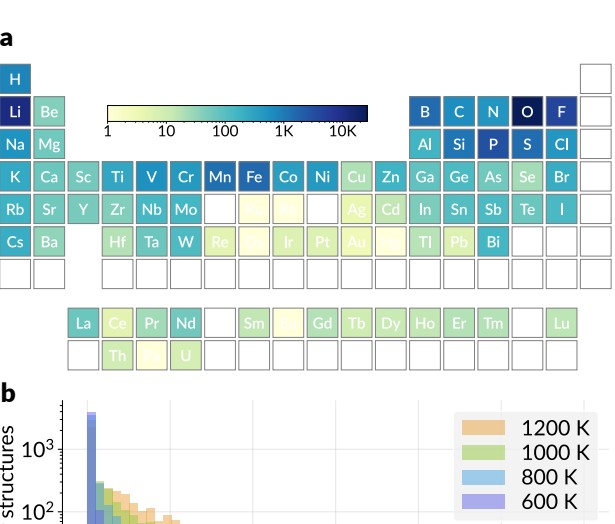

*Figure 2.* **Dataset statistics.** (a) Elemental count distribution across structures in the dataset. (b) Histogram of lithium MSD values from 25-ps MD simulations at different temperatures.

---

over the convex hull < 0.1 eV/atom. These criteria are designed to sample various modes of lithium ion dynamics

across different compositions, while maintaining minimal requirements for the solid-state electrolytes. After building a supercell of the structure in order to ensure that each dimension is larger than 9 Å and minimizing the structure, we conducted NVT MD simulations with MACE-MP-0 small model (Batatia et al., 2024) at 600, 800, 1000, and 1200 K for each structure. The initial velocities were assigned according to the temperature, and the system was propagated for 25 ps with the time step of 1 fs (25,000 steps) using Nosé–Hoover dynamics (Nosé, 1984; Hoover, 1985) as implemented in ASE (Larsen et al., 2017). We recorded the atom positions every ten steps.

The element distribution of the structures are shown in Fig. 2a, and it encompasses 77 elements over the periodic table. The lithium MSD per structure over the 25 ps trajectories are shown in Fig. 2b. The results indicate that the dataset encompasses a wide range of atomic environments and dynamics, including a sufficient amount of data points containing diffusive lithium atoms.

## 3.2. Experiment settings

**Dataset split.** We divided the structures based on their composition into training (90%) and testing (10%) sets. We conducted experiments using three types of splits.

1. Composition split: Trained on 800 K MD trajectories of training structures and compared the results with 800 K MD trajectories of testing structures.
2. Temperature split: Trained on MD trajectories of test structures at all temperatures except the one being tested (e.g., trained on 800, 1000, 1200 K and tested on 600 K) and compared with test MD trajectories at that $T$.
3. Composition + temperature split: Trained on training structures at all temperatures except the one being tested and compared with test MD trajectories at that $T$.

**Inference setup.** We conducted LiFLOW inference iteratively for $N_{\text{step}} = 25$ steps to simulate dynamics over 25 ps with a time step of $\Delta\tau = 1$ ps. Each inference step involves $N_{\text{flow}} = 10$ flow matching iterations of both *Propagator* and *Corrector* models, resulting in a total of 20 forward passes of the PaiNN model per LiFLOW step. During each inference process for a given structure, we terminated when either the maximum number of steps ($N_{\text{step}}$) was reached or the model prediction diverged due to instabilities. We then logged the number of stable propagation steps for each inference.

**Metrics.** To quantify the prediction of kinetic observables, we compared the MSD of lithium atoms over 25 ps to the trajectories, calculated as $\text{MSD}_{\text{Li}} = (1/N_{\text{Li}}) \sum_{i,a_i=\text{Li}} \|\mathbf{x}_{i,\tau} - \mathbf{x}_{i,0}\|^2$ where $N_{\text{Li}}$ is the number of lithium atoms. Since the raw MSD values span wide orders of magnitude, we

compared the log MSD values to the reference trajectories. We report the mean absolute error (MAE) and Spearman's rank correlation ($\rho$) for log MSD predictions, as well as the number of stable propagation steps in LiFLOW inference.

**Regression baseline.** We additionally trained a regression model for log MSD values with initial structure as inputs. We used the same PaiNN architecture as the *Propagator* and *Corrector* models, but switched the output layer to predict a single scalar for each node, which are then averaged to give a structure-level prediction.

## 3.3. Results

**Reproducing kinetic observables.** We report the test metrics on various splits in Table 1. Throughout our analyses, we found a consistent Spearman correlation around 0.7 across various split scenarios, except for the final split, which is a particularly challenging case. Given that the baseline model receives identical input (initial atom positions) and is directly trained to predict the log MSD values, it demonstrates better performance on composition splits and predictions at higher temperatures. Note that the regression baseline does not provide the dynamics of individual atoms. However, LiFLOW model exhibits superior performance in generalizing to lower temperature predictions. This difference in generalizability across different temperatures may stem from different magnitudes of prior displacements, which introduces complexity to the generation process at higher temperatures. Lower temperature predictions may be effectively learned from data collected at elevated temperatures, as the prior distribution at lower temperatures could potentially be encompassed within those of higher temperatures. This suggests that the model could leverage information from higher temperature data to inform its predictions at lower temperatures, offering a potential explanation for the observed performance disparity.

When trained on trajectories at identical temperatures, the model mostly overestimate the MSD, resulting in errors associated with identifying "false positive" fast lithium ion conductors (also refer to Fig. 3). Given that the true MSD values are relatively small in these cases (less than 1 Å$^2$), the corresponding structures typically manifest dynamic behavior wherein most lithium atoms remain within their original crystallographic sites, undergoing vibrations around their equilibrium positions. This outcome underscores the potential benefits of integrating pre-trained MLIP features into the model inputs. These features are inherently informative about the stability of the current configuration, suggesting that their incorporation could significantly enhance the prediction accuracy for low-MSD structures.

**Effect of the corrector model.** As evidenced by the average number of stable steps in Table 1 and illustrated in Fig. 3,

*Table 1.* **Prediction results.** Mean Absolute Error (MAE) and Spearman rank correlation ($\rho$) for the log MSD values (MSD in units of $Å^2$) are displayed for all models, while the average number of stable steps is reported for LiFLOW models. The best results for each split are highlighted in bold, with the best among LiFLOW models underlined if not bolded.

| Model | Composition split 800 K | Temperature split 600 K | Temperature split 1200 K | Comp. + temp. split 600 K | Comp. + temp. split 1200 K |
|---|---|---|---|---|---|
| REGRESSION PREDICTOR | logMSD MAE ($\downarrow$) / logMSD Spearman $\rho$ ($\uparrow$) | | | | |
| Structure input | **0.33 / 0.77** | 0.77 / 0.76 | 0.55 / **0.87** | 0.88 / 0.66 | **0.69 / 0.75** |
| LiFLOW | logMSD MAE ($\downarrow$) / logMSD Spearman $\rho$ ($\uparrow$) / Average num. stable steps ($\uparrow$) | | | | |
| *Propagator* only | 0.62 / 0.57 / 12.5 | 0.36 / 0.76 / 20.6 | 0.80 / 0.56 / 8.4 | 0.41 / 0.67 / 20.8 | 0.91 / 0.22 / 5.2 |
| *Propagator + Corrector* | 0.70 / 0.69 / 24.3 | **0.34 / 0.77** / 23.8 | **0.55** / 0.68 / 21.1 | **0.36 / 0.72** / 24.4 | 0.78 / 0.48 / 19.2 |

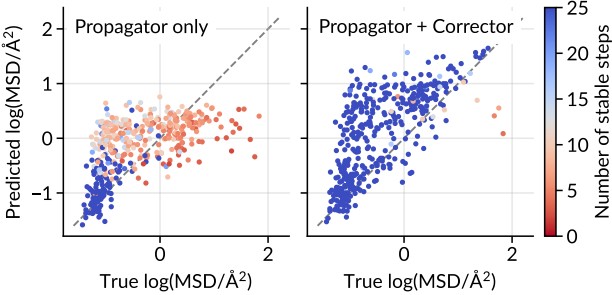

*Figure 3.* **Effect of the corrector model.** The scatter plot of the true and predicted log MSD values from the composition split experiment at 800 K, comparing the performance of the *Propagator*-only and *Propagator + Corrector* LiFLOW models.

we observed enhanced stability in propagation when utilizing the *Corrector* model. In contrast, the *Propagator*-only model struggled to propagate atomic positions over significant distances, resulting in log MSD values hovering below 1. The stochastic nature of the LiFLOW *Propagator* model necessitates a substantial dataset size to adequately cover the distribution of potential atomic movements over extended time intervals. However, since the data collection relies on MD simulations using MLIPs across diverse materials stuctures, it is challenging to gather a sufficiently large amount of data, as with the biomolecular simulations using classical force fields (e.g., Klein et al. (2024) and Schreiner et al. (2024)). Consequently, errors in *Propagator* predictions are inevitable, compounded by the autoregressive nature of inference, leading to divergence in propagation over time. The *Corrector* model addresses this issue by mapping erroneous atom positions after propagation to align with thermally plausible distributions, thereby stabilizing propagation and enabling longer simulation steps.

**Prediction speed.** Experiments were performed using a single NVIDIA RTX A5000 GPU. For a $2 \times 2 \times 1$ supercell of LGPS ($Li_{10}GeP_2S_{12}$, mp-696128) containing 200 atoms, a 25,000-step NVT MD simulation requires 530 seconds. In contrast, LiFLOW model inference with $N_{step} = 25$ com-

pletes in 1.8 seconds, resulting in approximately a $300\times$ speedup compared to MD simulation.

## 4. Conclusion

We proposed the LiFLOW model, a generative acceleration framework tailored for lithium-based solid-state electrolyte molecular dynamics (MD) simulations. The model is composed of *Propagator* and *Corrector* components, which utilizes a conditional flow matching scheme to predict atomic displacements for time propagation and denoising, respectively. Our model achieves Spearman's rank correlation of approximately 0.7 when reproducing mean squared displacement (MSD) values on compositionally and thermally split test structures. Remarkably, LiFLOW achieves a speedup of about $300\times$ compared to reference MD simulations with machine learning interatomic potentials (MLIPs). While our model exhibits a tendency to overestimate MSD values for non-diffusing structures, we aim to address this issue by incorporating features from pretrained MLIP models to develop energy-aware propagation models, enhancing the accuracy and robustness of our approach.

While the underlying assumption regarding the sufficient accuracy of electronic structure calculations and MLIP approximations generally holds, recent reports (Deng et al., 2024) indicate that MLIPs may smooth the potential energy landscape and lead to an overestimation of kinetic properties. Therefore, it's essential to consider the accuracy of the reference dynamics. However, the methodology employed in the current study is focused on rapidly screening potential electrolyte materials, prioritizing speed over absolute accuracy in generating dynamics. Beyond facilitating high-throughput virtual screening for electrolyte materials, we envision that based on an efficient and differentiable description of ionic transport, future application of our framework will enable inverse design and optimization of electrolyte materials.

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
