# OpenReview forum: "Generative acceleration of molecular dynamics simulations for solid-state electrolytes"
_ICML.cc/2024/Workshop/ML4LMS — ML4LMS Oral_

### Official Review · Reviewer_aJRe · 2024-06-03
**Generative acceleration of MD**

**Rating:** 8
**Confidence:** 3

**Review:**

The authors propose a generative conditional flow matching method to accelerate molecular dynamics simulations of solid-state electrolytes. Their framework uses two components, a propagator and a corrector, to predict atomic positions at timesteps much larger than the conventional timesteps used in MD. They intend to achieve temperature and chemical composition transferability, curating a data set for this purpose. The experiments are compelling, and the approach is well-motivated, in the path towards enabling a fast screening of candidate materials. They achieve a 2-order-of-magnitude speed-up compared to an MLIP, being the latter ones already much faster than AIMD. Given the positive results, I recommend its acceptance and I give it a good score (8/10).

---

### Official Review · Reviewer_Lb3w · 2024-06-10

**Rating:** 7
**Confidence:** 4

**Review:**

**Summary:**

The authors propose a method to accelerate dynamic simulations in solids, with particular application to diffusion in solid state electrolytes.
The approach is based on two flow matching modules working in the space of atomic coordinates, through a network architecture based on PaiNN.

**Pros:**

The approach is valid and integrates existing models in a novel way to tackle an important problem.

The results seem quite promising, although not perfect for all setups considered.

**Cons:**

The main result of the paper is somewhat undermined by the fact that the baseline regression predictor is able to match or surpass the generative approach in MSD prediction for most setups.
While the authors correctly point out that the diffusion model gives access to much more information about the system, no attempt is made to analyze it: it would be interesting to see whether the evolved structure are physically plausible or some other statistical property.
If the target property can be better estimated directly (which should be even faster than the diffusion model) and other physical properties are not correctly reproduced, there would be little incentive to use this method, in its current form.

Moreover, this work claims to build upon existing methods, but no effort is made to compare the results with any of those. While some of them tackle quite different systems, it would be useful to include some comments on whether they could be adapted to this sort of systems, or even better a comparison on some common target, also considering that some of them are shown to work on even longer timescales.

---

### Official Review · Reviewer_A45T · 2024-06-11
**The paper is of good quality and can be accepted**

**Rating:** 8
**Confidence:** 4

**Review:**

- typo line 039 am initio should be italics
- typo 043 spatio-temporal

The paper is well constructed and is relevant to the workshop proposing and highlighting the use of the corrector for de-noising purposes. However, the only problem I have with this paper is the data splitting, being 90% for training and 10% for testing. I wished they used a validation set too.